# Integrated Approaches to Invasive Fruit Fly Disinfestation: Ethyl Formate Fumigation and Cold Treatment for *Bactrocera scutellata* as a Surrogate in Korea

**DOI:** 10.3390/insects16070658

**Published:** 2025-06-24

**Authors:** Dongbin Kim, Tae Hyung Kwon, Bongsu Kim, Gi-Myeon Kwon, Sung-Eun Lee, Byung-Ho Lee

**Affiliations:** 1Department of Applied Biosciences, Kyungpook National University, Daegu 41566, Republic of Korea; dongbinkim@knu.ac.kr (D.K.); selpest@knu.ac.kr (S.-E.L.); 2Institute of Agricultural and Life Science, Gyeongsang National University, Jinju 52828, Republic of Korea; 3Oak Ridge Institute for Science and Education (ORISE), Oak Ridge, TN 37831, USA; teahyeong.kwon@usda.gov; 4Pacific Basin Agricultural Research Center, US Department of Agriculture-Agricultural Research Service, Hilo, HI 96720, USA; 5Plant Quarantine Technology Center, Animal and Plant Quarantine Agency (APQA), Gimcheon 39660, Republic of Korea; bskim79@korea.kr; 6Biological Utilization Institute, Andong 36729, Republic of Korea; scalekgm@hanmail.net; 7College of Environmental and Life Science, Murdoch University, Murdoch, WA 6150, Australia

**Keywords:** *Bactrocera scutellata*, fruit fly, ethyl formate, phytosanitary treatment

## Abstract

The oriental fruit fly, *Bactrocera dorsalis*, is a significant insect pest threatening mandarin production and trade. This study evaluated ethyl formate (EF)-based treatments combined with either cold treatment or phosphine (PH_3_) fumigation, using *B. scutellata* as a domestic fruit fly species. The combination of EF with cold treatment or PH_3_ enhanced the overall effectiveness, suggesting that EF-based integrated treatments could offer more practical and efficient quarantine methods against *B. dorsalis*.

## 1. Introductions

Global climate change and the expansion of international trade are accelerating the spread of invasive species, including quarantine insect pests, which in turn is leading to a decline in biodiversity and threatening ecosystem stability [1,2,3,4,5]. Rising temperatures and shifting precipitation patterns have facilitated the geographical expansion of many insect pests [6,7], while the diversification of cultivated crops has increased the availability of suitable host plants [8]. Consequently, the probability of invasive pest introduction and establishment has significantly increased, posing a serious threat to agricultural production and ecosystem stability [9]. Without effective monitoring and stringent phytosanitary measures, these environmental changes could result in severe economic and ecological consequences [6].

Among invasive quarantine pests, fruit flies represent a significant challenge due to their broad host range and adaptability, necessitating strict quarantine measures. Over the past 3 years, more than 50 cases of import prohibitions have been recorded due to detections of the Mediterranean fruit fly (*Ceratitis capitata*), Queensland fruit fly (*Bactrocera tryoni*), and Oriental fruit fly (*Bactrocera dorsalis*) in vineyards and orchards across Chile, Australia, and the United States, affecting the trade of key agricultural commodities such as grapes, kiwis, and oranges [10].

While phytosanitary disinfestation treatments are actively studied for invasive fruit fly species, research on quarantine-prohibited pests such as *C. capitata* and *B. dorsalis* is highly restricted in Korea. As a result, an alternative approach is necessary to evaluate effective disinfestation methods. Instead of directly studying *B. dorsalis*, which is not present in Korea, this study utilized the pumpkin fruit fly (*B. scutellata*), a native non-quarantine species with similar biological traits. This surrogate species was selected due to its frequent occurrence in Korea, ease of rearing, and relevance in pest management research. *B. scutellata* lays eggs in pumpkin flowers, and previous studies have investigated its local distribution and associated crop damage [11]. The current management strategies for *B. dorsalis* include the use of the male annihilation technique (MAT), which targets males by employing specific attractants such as methyl eugenol (ME) or raspberry ketone (RK) combined with insecticides such as spinosad. This method effectively reduces the male population, thereby suppressing the overall pest population. Additionally, control measures utilizing terpinyl acetate (TA) in conjunction with protein-based lures are implemented to attract and manage both male and female flies, enhancing the efficacy of integrated pest management programs [12,13], but its potential as a model organism for quarantine research remains underexplored.

Phytosanitary treatments are essential for mitigating the risk of introducing and establishing invasive pest species through international trade. In this context, the present study investigated the efficacy of fumigation and cold treatment protocols against *B*. *dorsalis*, using *B*. *scutellata* as a surrogate species. The efficacy of these treatments is expected to provide baseline data for controlling invasive fruit fly species in quarantine processes.

Ethyl formate (EF) fumigation has been extensively studied as a potential alternative to methyl bromide (MB) in quarantine applications [14]. EF is a naturally occurring compound found in grains, water, and fruits [15] and is recognized as safe, being registered as a food additive by the U.S. Food and Drug Administration [16]. EF undergoes rapid hydrolysis upon exposure to moisture, breaking down into ethanol and formic acid, thereby minimizing residue concerns [17]. EF fumigation has already been successfully applied to fruits [18,19,20,21,22,23,24,25], vegetables [26,27], and other agricultural commodities [28,29], making it a promising candidate for phytosanitary treatment.

Phosphine (PH_3_) is another widely used fumigant, particularly for stored grain insect pests, and has been explored as an alternative to MB in fruit and nursery plant treatments [30,31,32]. However, PH_3_ fumigation requires at least 24 h for effective control [31] and shows reduced efficacy at low temperatures [33]. Additionally, PH_3_ resistance has become widespread among stored grain pests, necessitating careful application to avoid resistance development [34,35].

Both fumigation and physical treatments, such as cold storage, are widely employed in pest management . The efficacy of cold treatment has been documented for multiple fruit fly species, including *Anastrepha suspensa* (Caribbean fruit fly), *B. dorsalis* (oriental fruit fly), and *C. capitata* (Mediterranean fruit fly) [36,37,38,39]. However, extended cold treatment durations pose logistical challenges and increase operational costs.

Given the need for alternative disinfestation methods for *B. dorsalis*, this study used *B. scutellata* to evaluate the efficacy of EF and PH_3_ fumigation, cold treatment, and their combination. The aim is to establish baseline models for invasive fruit fly control, providing essential data for developing future quarantine treatments against *B. dorsalis* and other high-risk quarantine insect pests.

## 2. Materials and Methods

### 2.1. Insect Rearing

The *B. scutellata* was reared at the Bio-Uilization Research Institute in Andong, Gyeongsangbuk-do, Republic of Korea. Adults were maintained in a two-tier cylindrical acrylic cage (Ø: 250 mm × height: 320 mm). To maintain a relative humidity of approximately 60%, water was supplied to the bottom section of the cage as needed. The adult diet consisted of a mixture of sugar, whole milk powder, and yeast in a ratio of 2:1:1, with water added at 0.5 parts. Additionally, 2% agar was incorporated to facilitate water supply.

For oviposition, pumpkin stems were cut into 2 cm segments, wrapped in cotton, and placed in vials. A shaded area was provided on top of the acrylic cage to offer resting space for the adults. Eggs were transferred to a rearing container (210 × 160 × 110 mm), where pumpkin flower pistils were provided as a food source. Upon pupation, the pupae were buried at a depth of 5 cm in a rearing container filled with fertilized soil. To prevent desiccation, water was supplied every 2–3 days.

Emerging adults were collected and reintroduced into the cylindrical acrylic cage for subsequent oviposition. The rearing conditions were maintained at 25 ± 3 °C, with a relative humidity of 60 ± 10%, under a photoperiod of 15L:7D.

### 2.2. Assessment of Fumigation and Cold Treatment Against Bactrocera scutellata

#### 2.2.1. Preparation: Naked Condition with 2% Agar Medium

Cold treatment experiments under naked conditions were only conducted on eggs and 3rd instar larvae of *B. scutellata* due to conspecific predation and cannibalism observed during the 1st–2nd instar larvae. For the egg stage experiment, eggs were extracted from pumpkin stems used for oviposition induction and placed in Petri dishes (Ø: 4.5 cm) containing a 2% agar medium, with at least 20 eggs per dish. Third-instar larvae were separated from the pumpkin flowers provided as food and similarly placed in Petri dishes (Ø: 4.5 cm) containing a 2% agar medium, with at least 20 larvae per dish. All experiments were conducted in three replicates for experimental reliability.

#### 2.2.2. Preparation: Inoculated Condition in Mandarin

The results of the cold treatment on *B. scutellata* under naked conditions showed that 3rd instar larvae exhibited a similar level of susceptibility to cold treatment as invasive fruit fly species, according to a reference study [40]. However, eggs demonstrated significantly different responses. Due to these differences in cold susceptibility, all following efficacy experiments under inoculated conditions were conducted on 3rd instar larvae. The 3rd instar larvae of *B. scutellata* were artificially inoculated into locally purchased mandarins, with at least 20 larvae inoculated per fruit. All experiments were conducted in three replicates for experimental reliability.

#### 2.2.3. Physical Treatment Process: Cold Treatment

Cold treatments were applied under both naked and inoculated conditions at 1.7 °C for varying periods using a temperature-controlled chamber. After treatment, the treated and untreated controls were transferred to a rearing room to assess egg hatchability after 3 days and the mortality of 3rd instar larvae after 3 days. All experiments were conducted in triplicate for experimental reliability.

#### 2.2.4. Chemical Treatment Process: Fumigation

A fumigation experiment was carried out under naked and inoculated conditions using ethyl formate (Fumate™) (99% liquid ethyl formate, SafeFume Co., Ltd., Daegu, Republic of Korea) and phosphine (Vivakill™) (2%, FarmHannong Co., Ltd., Seoul, Republic of Korea) and a glass desiccator (Duran^®^, 6.8 L, Duran Produktions GmbH & Co. KG, Mainz, Germany) for EF fumigation. To optimize the evaporation of liquid EF inside the desiccator, a piece of filter paper (Adventec MFS Inc., Grade 1, Dublin, CA, USA) was inserted into the glass stopper, serving as a surface for vaporization. Additionally, a magnetic bar was positioned at the bottom of the desiccator to ensure even distribution of the evaporated EF, and a magnetic stirrer was utilized to promote air circulation.

Fumigation was conducted at 20 °C for 4 h under both naked and inoculated conditions. After treatment, the treated and untreated controls were transferred to a rearing room to assess egg hatchability after 3 days and the mortality of 3rd instar larvae after 3 days. All experiments were conducted in triplicate for experimental reliability.

#### 2.2.5. Combination Treatment Process: Ethyl Formate with Phosphine Fumigation

To evaluate the potential of combined EF and PH_3_ treatment, 3rd instar larvae of *B. scutellata* were exposed to a combined treatment under naked conditions at 20 °C for 4 h. Based on the efficacy of EF, PH_3_ was applied at a concentration of 1.0 g/m^3^ in combination with EF at LCt_25%_ and LCt_50%_ levels, where lethal concentration x time (LCt) (indicating the product of Ct that causes 25% and 50% mortality, respectively) and mortality were assessed after 3 days. All experiments were conducted in triplicate for experimental reliability.

#### 2.2.6. Systemic Treatment Process: Ethyl Formate Following Cold Treatment

To evaluate the potential of systemic treatment, 3rd instar larvae of *B. scutellata* were exposed to cold treatment following EF fumigation under both naked and inoculated conditions. The fumigation was conducted at 20 °C for 4 h, followed by cold treatment. Based on the efficacy of EF, larvae were treated at LCt_50%_ for each inoculated condition, and mortality was assessed over different periods of cold treatment at 1.7 °C. All experiments were conducted in triplicate for experimental reliability.

### 2.3. Monitoring and Analysis: Fumigant Concentration and Ct Products

The concentration of EF within the fumigation chamber was determined by analyzing gas samples collected at 0.1, 1.0, 2.0, and 4.0 h after EF application. Gas samples were separated using a DB5-MS column (30 m × 0.25 mm i.d., 0.25 µm film thickness; J&W Scientific, Folsom, CA, USA) installed in a Shimadzu GC-17A (Shimadzu, Kyoto, Japan) equipped with a flame ionization detector (FID). The oven temperature was maintained at 100 °C, while the injector and detector temperatures were set at 250 °C and 280 °C, respectively. Helium was used as the carrier gas at a flow rate of 1.5 mL/min. The EF concentration was determined by comparing the peak area to a series of external EF standards.

For PH_3_ analysis, the DB5-MS column (30 m × 0.25 mm i.d., 0.25 µm film thickness; J&W Scientific, Folsom, CA, USA) and Shimadzu GC-17A (Shimadzu, Kyoto, Japan) were used, but the detector was a nitrogen–phosphorus detector (NPD). The oven and injector temperatures were set at 250 °C, while the detector temperature was maintained at 320 °C. Helium was used as the carrier gas at a flow rate of 1.5 mL/min.

A 1 L Tedlar^®^ bag (SKC Inc., Eighty-Four, PA, USA) was used to generate an EF and PH3 standard based on the ideal gas law. The standard was analyzed using GC-FID and GC-NPD to obtain peak values, which were subsequently used to derive a linear regression equation. To measure the EF and PH3 concentration inside the treated desiccator, gas samples were extracted with a gas-tight syringe and analyzed by GC-FID and GC-NPD. The derived linear regression equation was then applied to accurately determine the EF concentration within the desiccator.

The Ct product was calculated using the following equation:Ct= ∑ ((C_I + C_(i + 1))(t_(i + 1) + t_i))/2
where C represents EF concentration (g/m^3^), t denotes fumigation time (hours), and Ct is the cumulative concentration–time product (g h/m^3^).

### 2.4. Statistical Analysis

The efficacy of EF against *Batrocera scutellata* was analyzed using Probit analysis (SAS Ver. 9.4) based on lethal concentration × time (LCt) and lethal time (LT) values [41]. LCt_50%, 99%_ and LT_50%, 99%_ were calculated to determine the time required to kill 50% and 99% of the pests at a given concentration. The reliability of results was assessed through the regression slope, R², and confidence intervals. The LCt and LT values were then used to compare the susceptibility of different developmental stages of *B. scutellata* to the fumigant.

Mortality under combination treatment and systemic treatment was analyzed using ANOVA through the SAS program with Tukey’s Studentized Range (HSD) test applied to determine statistical significance among treatment groups.

## 3. Results

### 3.1. Physical Treatment Process: Cold Treatment

Cold treatment for *B. scutellata* was conducted at 1.7 °C using a temperature-controlled chamber. Mortality at different cold treatment periods was analyzed using Probit analysis to determine lethal time (LT) values. Under the naked condition, the LT_50%_ and LT_99%_ for eggs were 3.8 and 11.9 days, respectively. For third-instar larvae, LT_50%_ and LT_99%_ were 3.5 and 8.6 days, respectively. Under naked condition, eggs exhibited higher tolerance to cold stress compared to third-instar larvae. Under the inoculated condition, LT_50%_ and LT_99%_ were 3.5 and 12.4 days, indicating reduced efficacy compared to the naked condition (Table 1).

### 3.2. Chemical Treatment Process: Fumigation

EF fumigation treatment for *B. scutellata* was conducted at 20 °C for 4 h. Mortality based on EF Ct values was analyzed using Probit analysis to determine lethal concentration × time (LCt) values. Under the naked condition, LCt_50%_ and LCt_99%_ for eggs were 73.0 and 397.8 g h/m^3^, respectively. For third-instar larvae, LCt_50%_ and LCt_99%_ were 112.4 and 265.7 g h/m^3^, respectively. Under the naked condition, eggs exhibited higher tolerance to EF compared to third-instar larvae. Under the inoculated condition, LCt_50%_ and LCt_99%_ were 215.6 and 1111.0 g h/m^3^, indicating reduced efficacy compared to the naked condition (Table 2).

### 3.3. Combination Treatment Process: Ethyl Formate with Phosphine Fumigation

Based on the efficacy results of EF on third-instar larvae of *B. scutellata*, a combination treatment of EF LCt_25%_, LCt_50%_, and PH_3_ (1.0 g/m^3^) was conducted at 20 °C for 4 h. When treated with PH_3_ (1.0 g/m^3^) alone for 4 h, the mortality was 53.3 ± 1.7%. EF LCt_25%_ alone resulted in a mortality rate of 28.2 ± 1.7%, while the combination of EF LCt_25%_ + PH_3_ (1.0 g/m^3^) achieved a mortality of 91.7 ± 6.0%. EF LCt_50%_ alone resulted in 46.7 ± 1.7% mortality, whereas the combination of EF LCt_50%_ + PH_3_ (1.0 g/m^3^) achieved 100% mortality (Figure 1).

### 3.4. Systemic Treatment Process: Ethyl Formate Following Cold Treatment

Based on the efficacy of EF on third-instar larvae of *B. scutellata* under the naked condition, a systemic treatment was conducted by applying EF (LCt_50%_) at 20 °C for 4 h followed by cold treatment at 1.7 °C. In the cold treatment alone, mortality at 1, 3, and 5 days was 3.3, 30.0, and 86.7%, respectively, achieving 100% after 7 days. Under systemic treatment, mortality at 1, 3, and 5 days was 60.6, 66.7, and 90.0%, respectively, also achieving 100% after 7 days (Figure 2).

In contrast, under the inoculated condition, the efficacy of EF on third-instar larvae of *B. scutellata* was evaluated with systemic treatment. In the cold treatment alone, mortality at 1, 3, 5, 8, 10, and 12 days was 5.5, 10.0, 43.3, 79.7, 91.7, and 100%, respectively. Under systemic treatment, mortality at the same time points was 26.0, 27.4, 45.6, 88.9, 94.4, and 100%, respectively (Figure 3).

## 4. Discussion

With the expansion of tropical crop cultivation in Korea, concerns are increasing over potential host availability for invasive fruit flies [42]. This study aimed to assess the efficacy of various disinfestation methods using *B. scutellata* as a surrogate for *B. dosalis*, considering the anticipated introduction of invasive fruit flies due to climate change.

The results of the cold treatment (1.7 °C) indicated that *B. scutellata* eggs exhibited greater cold tolerance than third-instar larvae under naked conditions. This trend aligns with previous studies on *C. capitata* and *B. dorsalis*, where larvae showed greater tolerance to low temperatures [39]. However, the observed cold tolerance of *B. scutellata* eggs was notably higher than what has been reported for *B. dorsalis*, where eggs are generally more susceptible to cold exposure [40]. The difference in cold tolerance between these two species likely stems from their respective ecological and evolutionary backgrounds. *B. dorsalis* is a species native to tropical and subtropical climates and thus has not evolved significant cold tolerance mechanisms. As a result, its eggs are highly susceptible to cold stress, making cold treatment an effective quarantine measure. In contrast, *B. scutellata* is a temperate-zone species, so it has likely developed greater physiological resilience to cold exposure due to the seasonal temperature fluctuations in Korea. The ability of *B. scutellata* to overwinter in the egg stage suggests it has mechanisms to tolerate lower temperatures, which may explain the higher cold tolerance of its eggs compared to *B. dorsalis.*

In natural environments, organisms are concurrently exposed to multiple abiotic and biotic stressors that influence their survival, distribution, and adaptation [43]. One of the critical abiotic factors affecting insect survival is low temperature, which has driven the evolution of diverse physiological mechanisms to mitigate cold-induced damage. Insects exhibit two primary cold survival strategies: freeze tolerance and cold tolerance. Freeze-tolerant species can withstand internal ice formation by producing ice-nucleating agents (INAs) that facilitate extracellular freezing, thereby preventing lethal intracellular ice formation and subsequent cellular damage [44,45]. Additionally, these insects synthesize antifreeze proteins (AFPs), which reduce the freezing point of bodily fluids and minimize secondary damage during thawing [46].

In contrast, cold-tolerant insects do not experience freezing but rely on physiological regulation to endure prolonged exposure to low temperatures. Mechanisms such as diapause induction, protein stabilization, membrane fluidity adjustments, and stress response activation contribute to their survival, typically within a temperature range of 0–10 °C [47]. However, unlike freeze-tolerant species, cold-tolerant insects exist in a supercooled state to avoid ice crystallization, accumulating metabolic and cellular injuries over time, ultimately resulting in mortality. Understanding these cold tolerance strategies is essential for developing effective phytosanitary treatments targeting invasive insect pests. A comprehensive investigation of cold-induced regulatory pathways and adaptive molecular mechanisms is necessary to refine current pest management strategies [43,48]. Under inoculated conditions, the efficacy of cold treatment declined, likely due to the thermal buffering effect of mandarins, which delayed temperature penetration to lethal thresholds for the larvae. This observation underscores the importance of considering the thermal properties of host fruit when assessing cold treatment efficacy. Similar findings have been reported for C. capitata, where LT_99%_ values for eggs and early-instar larvae increased when infested inside grapes [40]. While *B. scutellata* serves as a useful surrogate species for evaluating cold disinfestation methods, its inherent cold tolerance differences from *B. dorsalis* necessitate caution when extrapolating these results to real quarantine applications. EF fumigation at 20 °C for 4 h revealed that third-instar larvae of *B. scutellata* were more susceptible than eggs, consistent with reports on *Planococcus citri, Pseudococcus comstocki*, and *Drosophila suzukii* [20,22,49]. However, EF efficacy declined significantly under inoculated conditions due to moisture-induced hydrolysis [17] and limited penetration through the mandarin’s physical structure. The high LCt_99%_ (1111.0 g h/m^3^) value under inoculated conditions suggests that EF fumigation alone may not be a practical quarantine treatment due to potential phytotoxic effects. EF-based treatments may require modifications, such as altered exposure times or concentration adjustments, to improve efficacy while maintaining fruit quality.

Combination treatments of EF and PH_3_ significantly enhanced mortality rates, with EF (LCt_50%_) + PH_3_ (1.0 g/m^3^) achieving 100% mortality within 4 h. These results align with previous studies demonstrating synergistic effects of EF-PH_3_ combinations for postharvest pest control in crops such as pineapples and citrus fruits [19,20]. Given PH_3_’s known limitations, such as long exposure requirements and increasing resistance among stored-product pests [34], further studies should explore optimized EF-PH_3_ dosage ratios to reduce treatment durations while maintaining effectiveness. Systemic treatment, where EF fumigation preceded cold exposure, demonstrated improved efficacy compared to cold treatment alone. Under naked conditions, mortality at 1.7 °C reached 100% within 7 days, whereas cold treatment alone required longer exposure times. Similar trends were observed under inoculated conditions, confirming previous research showing enhanced mortality when EF fumigation was followed by cold treatment in *Drosophila suzukii* [49].

Although statistical differences between systemic and cold treatments alone were minimal, systemic treatments could shorten quarantine durations, reducing logistical costs while maintaining efficacy. Further research should investigate optimal EF concentrations to balance efficacy and fruit quality preservation.

## 5. Conclusions

This study validated the use of *B. scutellata* as a surrogate species for evaluating phytosanitary treatment against *B. dorsalis*. In contrast to *B. dorsalis*, where eggs are generally more susceptible to cold, *B. scutellata* eggs exhibited higher cold tolerance than third-instar larvae. Therefore, while *B. scutellata* eggs may not be suitable surrogates for *B. dorsalis* due to their higher cold tolerance, the larvae of *B. scutellata* exhibited similar cold susceptibility to those of *B. dorsalis*, suggesting their potential use in surrogate-based studies.

While standalone EF treatment showed limitations, combination and systemic treatments demonstrated promise for improving fruit fly disinfestation strategies. Future research should focus on directly evaluating *B. dorsalis* in controlled quarantine research facilities such as BL3, in international collaborative works, and in direct studies on the target species to provide more precise baseline data for developing effective treatment protocols, ensuring the reliability of disinfestation strategies for global trade and phytosanitary regulations. Comparative assessments of cold tolerance mechanisms, mortality, and molecular responses between *B. scutellata* and *B. dorsalis* will be essential in refining treatment efficacy and optimizing quarantine measures.

## Figures and Tables

**Figure 1 insects-16-00658-f001:**
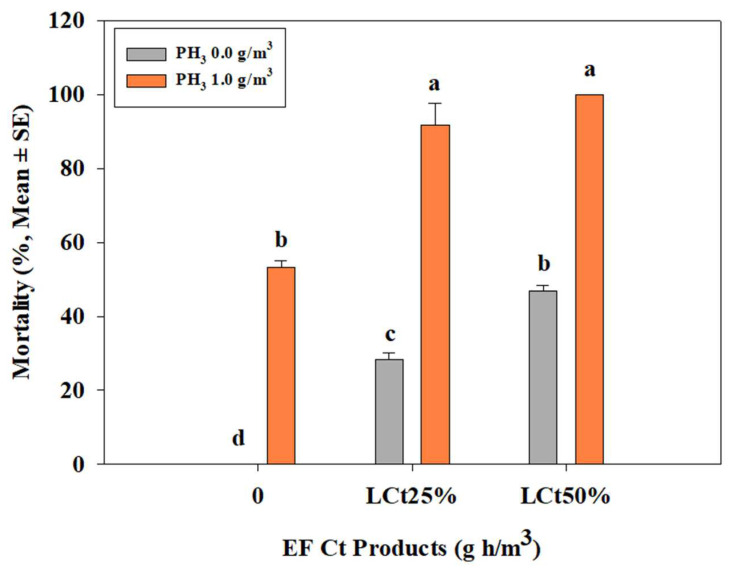
Evaluation of 4-h combined treatment with EF (LCt_25%, 50%_) and PH_3_ (1.0 g/m^3^) on 3rd instar larvae of *Bactrocera scutellata* under the naked condition (ANOVA: F(5, 12) = 252.60, *p* < 0.0001). Different letters indicate statistically significant differences.

**Figure 2 insects-16-00658-f002:**
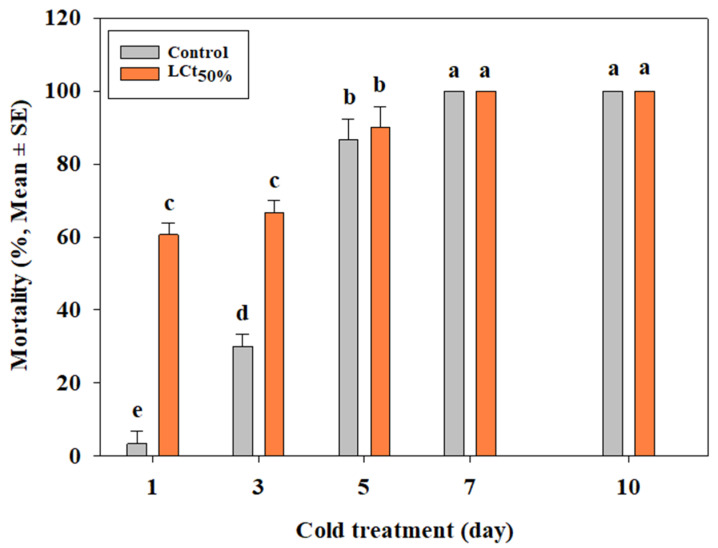
Evaluation of efficacy of systemic treatment (EF LCt_50%_ followed by cold treatment at 1.7 °C) on 3rd instar larvae of *Bactrocera scutellata* at 20 °C for 4 h under naked condition (ANOVA: F(9, 20) = 359.87, *p* < 0.0001). Different letters indicate statistically significant differences.

**Figure 3 insects-16-00658-f003:**
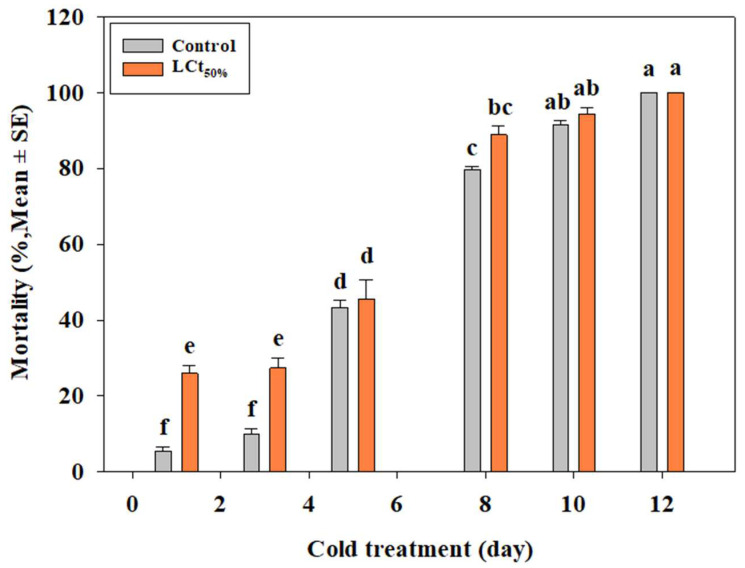
Evaluation of efficacy of systemic treatment (EF LCt_50%_ followed by cold treatment at 1.7 °C) on 3rd instar larvae of *Bactrocera scutellata* at 20 °C for 4 h under inoculated condition (ANOVA: F(11, 24) = 669.43, *p* < 0.0001). Different letters indicate statistically significant differences.

**Table 1 insects-16-00658-t001:** Evaluation of cold treatment (1.7 ± 0.5 °C) efficacy on different developmental stages of *Bactrocera scutellata*.

Stage	Condition	LT_50%_ (95% CI)	LT_99%_ (95% CI)	Slope ± SE	*df*	χ^2^
Egg	Naked	3.8 (3.0–4.5)	11.9 (8.8–21.4)	4.7 ± 0.8	28	67.2
Third-instar larvae	Naked	3.5 (2.6–4.4)	8.6 (6.3–19.0)	6.0 ± 1.5	34	23.4
	Inoculated	3.5 (2.5–4.5)	12.4 (8.5–28.7)	4.3 ± 0.8	16	25.17

**Table 2 insects-16-00658-t002:** Evaluation of 4-h EF fumigation on different developmental stages of *Bactrocera scutellata at* 20.0 ± 1.0 °C.

Stage	Condition	LCt_25%_ (95% CL)	LCt_50%_ (95% CL)	LCt_99%_ (95% CL)	Slope ± SE	*df*	χ^2^
Eggs	Naked	44.6 (32.0–56.2)	73.0 (58.3–86.8)	397.8 (303.1–597.3)	3.2 ± 0.4	28	35.8
Third-instar larvae		87.6 (76.3–96.4)	112.4 (102.8–122.8)	265.7 (217.7–373.0)	6.2 ± 0.9	16	12.8
Third-instar larvae	Inoculated	156.7 (130.5–164.1)	215.6 (174.5–253.0)	1111.0 (806.3–1911)	3.3 ± 0.5	16	49.38

## Data Availability

All the data generated in this work are provided in the article.

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
