# Peer review of "Integrated Approaches to Invasive Fruit Fly Disinfestation: Ethyl Formate Fumigation and Cold Treatment for *Bactrocera scutellata* as a Surrogate in Korea"

_insects, 2025, doi:10.3390/insects16070658_

Round 1
Reviewer 1 Report
Comments and Suggestions for Authors
- In the abstract, please refrain from using abbreviations.
- I’m not convinced that Bactrocera scutellata can serve as an alternative model for managing dorsalis due to the significant differences in their biology. Therefore, I suggest focusing on the specific species being studied. In the conclusion, you might consider introducing B. scutellata as a potential surrogate for B. dorsalis.
- In my opinion, the integration of chemical with cold treatment is a viable approach. However, I would appreciate your rationale for the combination of EF and phosphine.
- Line 131: According to reference?????
- Line 163: You are expected to explain the abbreviation, such as "LCt," the first time it is mentioned.
- The methodology, particularly in “Monitoring and analysis,” needs citation.
- Please refer to “B. scutellata” rather than “Batrocera scutellata.”
- Please remove lines 211-213.
- Line 227: “LCt (Lethal Concentration × time) values” should come earlier. I am inquiring about the value of this combination. Could you please explain how you integrated these two units? It appears that there is only one treatment time: “Evaluation of EF 4-hour fumigation.”
- Line 242: results of EF LCt25% maybe shown in the Table 2.
- Please consider removing the initial paragraph of the discussion section or integrating it with the introduction for a more cohesive presentation.
- Line 272: It is important to note that susceptibility to cold treatment differs from cold tolerance. A key point to discuss is that the LT50% values for both eggs and larvae are nearly identical; however, the LT90% value for larvae is lower than that of the eggs. Why? Please discuss.
- Line 282: The difference in cold susceptibility between these two species is not the focus of your research.
- Based on your findings, you are expected to explore the potential reasons for the following outcomes: first, the combination of EF and PH₃ achieving a 100% success rate; second, the synergistic effects of EF and cold treatment; and third, the variations in mortality rates between the inoculated and non-inoculated conditions. It is important to note that simply citing previous works is not enough.
- The content from lines 293 to 312 does not pertain to your research.
- Line 374: I don’t agree with “This study validated the use of Bactrocera scutellata as a surrogate species for evaluating phytosanitary treatment against Bactrocera dorsalis.
- A well-crafted conclusion is highly appreciated.
A minor correction is suggested
Author Response
Line 131: According to reference????
- Lima, C. F. D.; Jessup, A.; Mansfield, E. R.; Daniels, D. Cold treatment of table grapes infested with Mediterranean fruit fly Ceratitis capitata (Widemann) and Queensland fruit fly Bactrocera tryoni (Froggatt) Diptera: Tephritidae). N. Z. J. Crop Hortic. Sci. 2011, 39(2), 95-105.
Line 163: You are expected to explain the abbreviation, such as “LCt”, the first time it is mentioned.
- This has been checked and revised as suggested
- EF at LCt25% and LCt50% levels, where LCt (Lethal Concentration x time, indicating the product of Ct that causes 25% and 50% mortality, respectively), and mortality was assessed after 3 days.
The Methodology, particularly in “Mornitoring and analysis,” needs citiation.
- Lee, J. ; Kim, H. K.; Kyung, Y. J.; Park, G. H.; Lee, B. H.; Yang, J. O.; Koo, H. N.; Kim, G. H. Fumigation activity of ethyl formate and phosphine against Tetranychus urticae (Acari: Tetranychidae) on imported sweet pumpkin. J. Econ. Entomol. 2018, 111(4), 16255-1632.
Please refer to “B. scutellate” rather than “Bactrocera scutellat”
- This has been checked and revised as suggested
Please remove lines 211-213. “Cold treatment for Bactrocera scutellata was conducted at 1.7 °C using temperature-controlled chamber. Mortality at different cold treatment periods were analyzed using Probit analysis to determine LT (Lethal Time) values.”
- This has been checked and revised as suggested
Line 227: “LCt (Lethal Concentration x time) values” should come earlier. I am requiring about the value of the combination. Could you please explain how you integrated these two units? It appears that there is only one treatment: “Evaluation of EF 4-hour fumigation”.
- The LCt (Lethal Concentration × time) values were previously described in the Materials and Methods section.
- By fixing the exposure time at 4 hours, various concentrations were applied to determine the corresponding Ct values. Based on the resulting mortality, LCt values for the 4-hour exposure were established.
Line 242: Result of EF LCt25% maybe shown in the Table 2.
- This has been checked and revised as suggested
Line 272: It is important to note that susceptibility to cold treatment differs from cold tolerance. A key point to discuss is that the LT50% values for both eggs and larvae are nearly identical, however, the LT90% value for larvae is lower than that of the eggs. Why? Please discuss.
- Research on cold treatment for scutellata remains limited. As discussed, this species is presumed to possess cold tolerance due to its ability to overwinter. Nevertheless, the Probit analysis based on mortality from cold treatment was conducted accurately and confirmed its reliability.
Line 282: The difference in cold susceptibility between these two species is not the focus of your research.
- The comparative analysis was provided based on the consideration of scutellata as a potential surrogate for B. dorsalis.
Based on your findings, you are expected to explore the potential reasons for the following outcomes.: first, the combination of EF and PH3 achieving a 100% success rate; second, the synergistic effects of EF and cold treatment; and third, the variations in mortality rates between the inoculated and non-inoculated conditions. It is important to note that simply citing previous works in not enough.
- I will assess how the citations relate to my own research, and also provide a critical interpretation or analysis.
The content from lines 293 to 312 does not pertain to your research.
- Since the study focused on cold treatment, it was mentioned to specify the mechanism by which low temperatures affect insects. Additionally, in the context of winter in Korea, the discussion also includes how scutellata has developed resistance to low temperatures during the winter season.
Line 374: I don’t agree with “This study validated the use of Bactrocera scutellata as a surrogate species for evaluating phytosanitary treatment against Bactrocera dorsalis”. A well-crafted conclusion is highly appreciated.
- This has been checked and revised as suggested
- This study validated the use of scutellata as a surrogate species for evaluating phytosanitary treatments against B. dorsalis. In contrast to B. dorsalis, where eggs are generally more susceptible to cold, B. scutellata eggs exhibited higher cold tolerance than third instar larvae. Therefore, although B. scutellata eggs may not be suitable surrogates for B. dorsalis due to their greater cold tolerance, the larvae of B. scutellata exhibited similar cold susceptibility to those of B. dorsalis, supporting their potential use in surrogate-based studies.
Reviewer 2 Report
Comments and Suggestions for Authors
The manuscript entitled "Integrated Approaches to Disinfestation of Invasive Fruit Flies: Ethyl Formate Fumigation and Cold Treatment for Bactrocera scutellata as a Surrogate in Korea" is an interesting study, but the current version needs major revision.
Please use scientific style for writing your research article.
Further explain what a surrogate species is.
Make a sequence in writing the introduction.
The methodology section is unclear.
Statistics need clarification.
Results need more details regarding post analysis and with proper justification.
Discussion lacks previous results and reasoning. And please mention the italicized name in abbreviation instead of the full name.

Please revise your English language from native or expert.
Author Response
Line 29 : “Ethyl formate and Phospine” Why you selected these two compounds?
- This study offers foundational data to optimize EF-based quarantine treatments against dorsalis, supporting shorter treatment times and more cost-effective quarantine practices. Future studies should validate these findings under practical field conditions.
Line 46: “[1]” Need more than one references
- Bebber, D.P.; Marriott, F.H.C.; Gaston, K.J.; Harris, S.A.; Scotland, R.W. Crop pests and pathogens move polewards in a warming world. Clim. Chang. 2013, 3, 985-988.
- Skendžić, S.; Zovko, M.; Živković, I.P.; Lešić, V.; Lemić, D. The impact of climate change on Agricultural insect pest. Insects. 2021, 12(5), 440.
- Jung, J.M.; Lee, W.H.; Jung, S.H. Insect distribution in response to climate change based on a model: review of function and use of CLIMEX. Res. 2016, 46, 223-235.
- Wang, C.J.; Wang, R.; Yu, C.M.; Dang, X.P.; Sun, W.G.; Li, Q.F.; Wang, X.T.; Wan, J.Z. Risk assessment of insect pest expansion in alpine ecosystem under climate change. Pest Manag. Sci. 2021, 77, 3165-3178.
- Lima, C. F. D.; Jessup, A.; Mansfield, E. R.; Daniels, D. Cold treatment of table grapes infested with Mediterranean fruit fly Ceratitis capitata (Widemann) and Queensland fruit fly Bactrocera tryoni (Froggatt) Diptera: Tephritidae). N. Z. J. Crop Hortic. Sci. 2011, 39(2), 95-105.
- [41] Finney, D. J. Probit Analysis, 3rd; Cambridge University Press, UK, 1971.
- The efficacy of EF against Batrocera scutellata was analyzed using Probit analysis (SAS Ver. 9.4) based on LCt (Lethal Concentration × time) and LT (Lethal Time) values [41]. LCt50%, 99% and LT50%, 99% were calculated to determine the time required to kill 50% and 99% of the pests at a given concentration. The reliability of results was assessed through the regression slope, R², and confidence intervals. The LCt and LT values were then used to compare the susceptibility of different developmental stages of Bactrocera scutellata to the fumigant. Mortality under combination treatment and systemic treatment were analyzed using ANOVA through the SAS program with Tekey’s Studentized Range (HSD) test applied to determine statistical significance among treatment groups.

Round 2
Reviewer 1 Report
Comments and Suggestions for Authors
As the authors have not addressed my concerns regarding the discussion, I maintain my previous reservations.
Author Response
Thank you for nice comments
Reviewer 2 Report
Comments and Suggestions for Authors
Well revised
Author Response
Thank you for nice comments